# Neural encoding of task-dependent errors during adaptive learning

Chang-Hao Kao[1]\*, Sangil Lee[1], Joshua I Gold[2], Joseph W Kable[1]\*

[1]Department of Psychology, University of Pennsylvania, Philadelphia, United States; [2]Department of Neuroscience, University of Pennsylvania, Philadelphia, United States

**Abstract** Effective learning requires using errors in a task-dependent manner, for example adjusting to errors that result from unpredicted environmental changes but ignoring errors that result from environmental stochasticity. Where and how the brain represents errors in a task-dependent manner and uses them to guide behavior are not well understood. We imaged the brains of human participants performing a predictive-inference task with two conditions that had different sources of errors. Their performance was sensitive to this difference, including more choice switches after fundamental changes versus stochastic fluctuations in reward contingencies. Using multi-voxel pattern classification, we identified task-dependent representations of error magnitude and past errors in posterior parietal cortex. These representations were distinct from representations of the resulting behavioral adjustments in dorsomedial frontal, anterior cingulate, and orbitofrontal cortex. The results provide new insights into how the human brain represents errors in a task-dependent manner and guides subsequent adaptive behavior.

## Introduction

Errors often drive adaptive adjustments in beliefs that inform behaviors that maximize positive outcomes and minimize negative ones (*Sutton and Barto, 1998*). A major challenge to error-driven learning in uncertain and dynamic environments is that errors can arise from different sources that have different implications for learning. For example, a bad experience at a restaurant that recently hired a new chef might lead you to update your belief about the quality of the restaurant, whereas a similar experience at a well-known restaurant with a chef that has long been your favorite might be written off as a one-time bad night. That is, the same errors should be interpreted differently in different contexts. In general, errors that represent fundamental changes in the environment or that occur during periods of uncertainty should probably lead you to update your beliefs and change your behavior, whereas those that result from environmental stochasticity are likely better ignored (*d'Acremont and Bossaerts, 2016*; *Li et al., 2019*; *Nassar et al., 2019a*; *O'Reilly et al., 2013*).

Neural representations of key features of these kinds of dynamic, error-driven learning processes have been identified in several brain regions. For example, several studies focused on variables derived from normative models that describe the degree to which individuals should dynamically adjust their beliefs in response to error feedback under different task conditions, including the probability that a fundamental change in the environment just occurred (change-point probability, or CPP, which is a form of surprise) and the reducible uncertainty associated with estimates of environmental features (relative uncertainty, or RU). Correlates of these variables have been identified in dorsomedial frontal (DMFC) and dorsolateral prefrontal (DLPFC) cortex and medial and lateral posterior parietal cortex (PPC) (*Behrens et al., 2007*; *McGuire et al., 2014*; *Nassar et al., 2019a*). These and other studies also suggest specific roles for these different brain regions in error-driving learning, including representations of surprise induced by either state changes or outliers (irrelevant to state changes) in the PPC that suggest a role in error monitoring (*Nassar et al., 2019a*;

\*For correspondence:
chakao@sas.upenn.edu (C-HK);
kable@psych.upenn.edu (JWK)

*O'Reilly et al., 2013*), and representations of variables more closely related to belief and behavior updating in the prefrontal cortex (PFC) (*McGuire et al., 2014*; *O'Reilly et al., 2013*). However, these previous studies, which typically used continuous rather than discrete feedback, were not designed to identify neural signals related to a key aspect of flexible learning in uncertain and dynamic environments: responding to the same kinds of errors differently in different conditions.

To identify such task-dependent neural responses to errors, we adapted a paradigm from our previous single-unit recording study (*Li et al., 2019*). In this paradigm, we generated two different dynamic environments by varying the amount of noise and the frequency that change-points occur (i.e. hazard rate; *Behrens et al., 2007*; *Glaze et al., 2015*; *Nassar et al., 2012*; *Nassar et al., 2010*). In one environment, noise was absent and the hazard rate was high, and thus errors unambiguously signaled a change in state. We refer to this high-hazard/low-noise condition as the *unstable* environment, because most errors can be attributed to volatility. In another environment, noise was high and the hazard rate was low, and thus small errors were ambiguous and could indicate either a change in state or noise. We refer to this low-hazard/high-noise condition as the *noisy* environment, because most errors can be attributed to stochasticity. Thus, effective learning requires treating errors in the two conditions differently, including adjusting immediately to errors in the unstable environment but using the size of errors and recent error history as cues to aid interpretation of ambiguous errors in the noisy environment.

In our previous study, we found many single neurons in the anterior cingulate cortex (ACC) or posterior cingulate cortex (PCC) that responded to errors or the current condition, but we found little evidence that single neurons in these regions combined this information in a task-dependent manner to discriminate the source of errors or drive behavior. In the current study, we used whole-brain fMRI and multi-voxel pattern classification to identify task-dependent neural responses to errors and activity predictive of behavioral updating in the human brain. The results show task-dependent encoding of error magnitude and past errors in PPC and encoding of behavioral shifts in frontal regions including ACC, DMFC, DLPFC and orbitofrontal cortex (OFC), which provide new insights into the distinct roles these brain regions play in representing errors in a task-dependent manner and using errors to guide adaptive behavior.

## Results

Sixteen human participants performed a predictive-inference task (*Figure 1A*) while fMRI was used to measure their blood-oxygenation-level-dependent (BOLD) brain activity. The task required them to predict the location of a single rewarded target from a circular array of ten targets. The location of the rewarded target was sampled from a distribution based on the location of the current best target and the noise level in the current condition. In addition, the location of the best target could change according to a particular, fixed hazard rate (*H*). Two conditions with different noise levels and hazard rates were conducted in separate runs. In the noisy condition (*Figure 1B–C*), the rewarded target would appear in one of the five locations relative to the location of the current best target, and the hazard rate was low ($H = 0.02$). In the unstable condition (*Figure 1D–E*), the rewarded target always appeared at the location of the best target, and the hazard rate was high ($H = 0.35$). On each trial, participants made a prediction by looking at a particular target, and then were given explicit, visual feedback about their chosen target and the rewarded target. Effective performance required them to use this feedback in a flexible and task-dependent manner, including typically ignoring small errors in the noisy condition but responding to small errors in the unstable condition by updating their beliefs about the best-target location.

### Behavior

Nearly all of the participants' choice patterns were consistent with a flexible, task-dependent learning process (closed symbols in *Figure 2*). On average, they learned the location of the best target after a change in its location more quickly and reliably in the unstable than the noisy condition (*Figure 2A*). This flexible learning process had two key signatures. First, target switches (i.e. predicting a different target than on the previous trial) tended to follow errors of any magnitude in the unstable condition but only errors of high magnitude (i.e. when the chosen target was 3, 4, or five targets away from the rewarded target) in the noisy condition (sign test for $H_0$: equal probability of switching for the two conditions; error magnitude of 1: median = −0.35, interquartile range (IQR) =

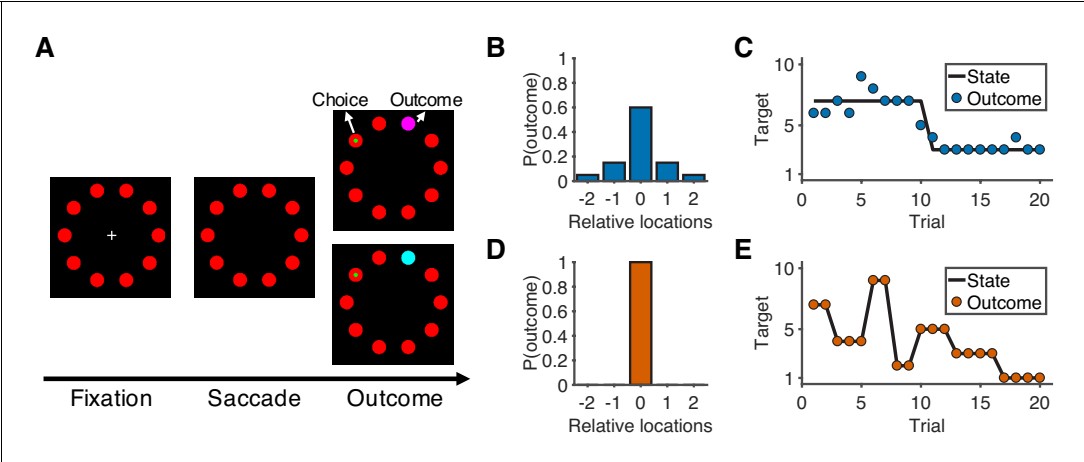

**Figure 1.** Overview of task and experimental design. (**A**) Sequence of the task. At the start of the trial, participants look at a cross in the center of the screen and maintain fixation for 0.5 s to initialize the trial. After the cross disappears, participants choose one of 10 targets (red) by looking at it within 1.5 s and then holding fixation on the chosen target for 0.3 s. During the outcome phase (1 s), a green dot inside the target indicates the participants' choice. The rewarded target is shown in purple or cyan to indicate the number of earnable points as 10 or 20, respectively. (**B**) Probability distribution of the rewarded target location in the noisy condition. Target location is relative to the location of the state (generative mean). The rewarded target probabilities for the relative locations of [−2,–1, 0, 1, 2] are [0.05, 0.15, 0.6, 0.15, 0.05]. (**C**) Example of trials in the noisy condition. The states change occasionally with a hazard rate of 0.02. (**D**) Probability distribution of the rewarded target location in the unstable condition. Because there is no noise in this condition, the rewarded target is always at the location of the state. (**E**) Example of trials in the unstable condition. The states change frequently with a hazard rate of 0.35.

[−0.62,–0.25], p<0.001; error magnitude of 2: median = −0.30, IQR = [−0.70,–0.11], p<0.001; *Figure 2B–C*). Second, target switches depended on error history only for low-magnitude errors (i.e. when the chosen target was 1 or two targets away from the rewarded target) in the noisy condition but not otherwise (sign test for $H_0$: switching was unaffected when recent history contained fewer errors; error magnitude of 1: median = −0.29, IQR = [−0.42,–0.10], p=0.004; error magnitude of 2: median = −0.25, IQR = [−0.38,–0.14], p<0.001; *Figure 2D–F*).

We accounted for these behavioral patterns with a reduced Bayesian model that is similar to ones we have used previously to model belief updating in a dynamic environment (open symbols in *Figure 2*; *Tables 1* and *2*). This model provides a framework to interpret and use errors differently according to the current task conditions, as defined by hazard rate and noise level. The decision-maker's trial-by-trial updates are governed by ongoing estimates of the probability that the best target changed (change-point probability, or CPP) and reducible uncertainty about the best target's location (relative uncertainty, or RU). Both quantities are influenced by the two free parameters in the model, subjective estimates of the task hazard rate and noise level, which were fitted separately in each condition for each participant. As expected, the fitted hazard rates were higher in the unstable condition than in the noisy condition, although both tended to be higher than the objective values, as we have observed previously (*Nassar et al., 2010*). However, the fitted noise estimates were not reliably different between the noisy and unstable conditions (*Table 2*). As we observed in our previous study (*Li et al., 2019*), the subjective estimates of noise level were high in the unstable condition despite the objective absence of noise.

We also tested several alternative models but they did not provide as parsimonious descriptions of the data (*Figure 2—figure supplement 2*, and *Tables 1* and *2*). Notably, an alternative model that assumed a condition-specific fixed learning rate also assumed errors were treated differently in the two conditions but did not include trial-by-trial adjustments of learning rates used by the reduced Bayesian model. Although this model performed better than the reduced Bayesian model in the unstable condition, it cannot capture participants' behaviors in the noisy condition, where dynamically integrating both current and past errors is required for adapting trial-by-trial behavior. Other hybrid models performed worse than the reduced Bayesian model in both conditions.

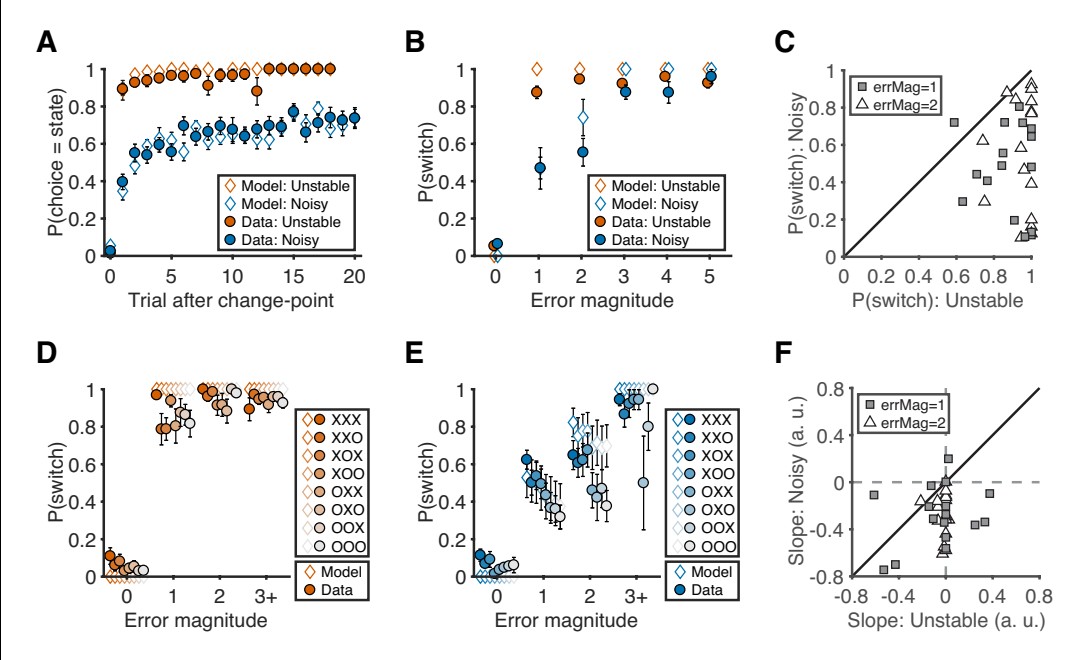

**Figure 2.** Behavioral results. (**A**) Probability of choosing the best target after change-points. Symbols and error bars are mean ± SEM across subjects (solid symbols) or simulations (open symbols). (**B**) Relationship between error magnitude and switch probability. Symbols and error bars are as in **A**. (**C**) The distribution of switch probabilities for small errors (magnitude of 1 or 2) in both conditions. Each data point represents one participant. Distributions for all error magnitudes are shown in *Figure 2—figure supplement 1*. (**D**) Probability of switch as a function of current error magnitude and error history in the unstable condition. Different colors represent different error histories for the past three trials. A correct trial is marked as O, and an error trial is marked as X. For example, XOO implies that trial t-1 was an error trial, and trial t-2 and trial t-3 were correct trials. Symbols and error bars are mean ± SEM across subjects. (**E**) Probability of switch as a function of current error magnitude and error history in the noisy condition. Symbols and error bars are as in **D**. (**F**) The distribution of the slopes of switch probability against error history for small errors (magnitude of 1 or 2) in both conditions. Each data point represents one participant. Distributions for all error magnitudes are shown in *Figure 2—figure supplement 1*.

The online version of this article includes the following figure supplement(s) for figure 2:

**Figure supplement 1.** Distributions of behavior as a function of error magnitude.
**Figure supplement 2.** Behavioral data and predictions from different models.
**Figure supplement 3.** Reduced Bayesian model applied to behavioral and imaging data.
**Figure supplement 4.** Neural representations of CPP and RU from the approximately ideal observer, which is the reduced Bayesian model with true hazard rate and noise, for direct comparison to analyses in *McGuire et al., 2014*, which used covariates constructed from the ideal rather than the fitted model.

## Neural representation of CPP and RU

The two key internal quantities in the reduced Bayesian model are CPP and RU, both of which contribute to processing errors in a task-dependent manner (*Figure 2—figure supplement 3*). CPP increases as the current error magnitude increases and achieves large values more quickly in the unstable condition because of the higher hazard rate. These dynamics lead to a greater probability of switching targets after smaller errors in the unstable condition. RU increases on the next trial after the participant makes an error and does so more strongly in the noisy condition because of the lower hazard rate. These dynamics lead to a greater probability of target switches when the last trial was an error, which is most prominent for small errors in the noisy condition. Thus, CPP and RU each account for one of the two key signatures of task-dependent learning that we identified in participants' behavior, with CPP driving a task-dependent influence of error magnitude and RU driving a task-dependent influence of error history on target switches.

Though not the main focus of this study, we were able to replicate our previous findings regarding the neural representations of CPP and RU (*McGuire et al., 2014*). Similar to our previous study,

**Table 1.** BIC of behavior models.

| Model | Condition | BIC improvement by RB model |
|---|---|---|
| Reduced Bayesian model (RB) | Unstable | |
| | Noisy | |
| Fixed learning rate model (fixedLR) | Unstable | 5.06 [3.63, 5.71]** |
| | Noisy | −21.05 [-76.63, 0.20]$^\dagger$ |
| RB + fixedLR | Unstable | −9.83 [-11.20,–8.07]*** |
| | Noisy | −4.64 [-10.51, 0.89] |
| RB + P$_{stay}$ | Unstable | −5.20 [-5.65,–3.68]** |
| | Noisy | −5.55 [-5.65,–2.67]* |

Values are shown as median [IQR]. A negative value means that the RB model performed better than the alternative model. Significance was tested by a sign test. $^\dagger p<0.08$, $**p<0.01$, $***p<0.001$.

we found activity that was positively correlated with the levels of CPP and RU across DLPFC and PPC (*Figure 2—figure supplements 3* and *4*). The regions of DLPFC and PPC that were responsive to both CPP and RU were a subset of those identified as showing this conjunction in our previous study. Because CPP and RU both contribute to responding to errors in a task-dependent manner, the brain regions that responded to both variables are good candidates for encoding errors in a task-dependent manner. In the following analyses, we aimed to directly identify task-dependent neural representations of error magnitude and error history, as well as activity that predicts subsequent shifts in behavior.

## Task-dependent neural representation of errors

We used multi-voxel pattern analysis (MVPA) to identify error-related neural signals that were similar and different for the two task conditions. Given the two key signatures of flexible learning that we identified in behavior, we were especially interested in identifying neural representations of error magnitude and past errors that were stronger in the noisy than the unstable condition.

We found robust, task-dependent representations of the magnitude of the error on the current trial in PPC. Consistent with the task-dependent behavioral effects, this representation of error magnitude was stronger in the noisy than the unstable condition (*Figure 3* and *Table 3*). Specifically, we could classify correct versus error feedback on the current trial across almost the entire cortex, in both the unstable and noisy conditions. However, for error trials, we could classify error magnitude (in three bins: 1, 2, 3+ targets away from the rewarded target) only for the noisy condition and most strongly in the lateral and medial parietal cortex and in the occipital pole. In a parallel set of

**Table 2.** Parameters of behavior models.

| Model | Parameter | Unstable | Noisy | Unstable > Noisy |
|---|---|---|---|---|
| RB | H | 0.82 [0.64, 0.90] | 0.33 [0.11, 0.50] | 0.37 [0.24, 0.62]*** |
| | K | 0.59 [0.03, 2.22] | 1.86 [1.22, 2.32] | −0.23 [-1.97, 0.71] |
| fixedLR | $\alpha_{fixed}$ | 0.96 [0.86, 0.97] | 0.63 [0.37, 0.73] | 0.33 [0.19, 0.49]*** |
| RB + fixedLR | H | 0.07 [0.00, 0.86] | 0.03 [0.00, 0.19] | 0.03 [-0.03, 0.77] |
| | K | 11.19 [2.78, 18.01] | 3.22 [2.28, 9.90] | 5.13 [-4.91, 16.10] |
| | $\alpha_{fixed}$ | 0.96 [0.75, 1.00] | 0.88 [0.23, 1.00] | 0.02 [-0.12, 0.52] |
| | w | 0.38 [0.16, 0.81] | 0.71 [0.52, 0.87] | −0.28 [-0.57, 0.22] |
| RB + P$_{stay}$ | H | 0.73 [0.64, 0.88] | 0.31 [0.06, 0.53] | 0.27 [0.15, 0.66]** |
| | K | 8.42 [0.73, 30.42] | 2.19 [1.62, 9.09] | 2.71 [-2.60, 23.94] |
| | $P_{stay}$ | 0.01 [0.00, 0.05] | 0.01 [0.00, 0.13] | 0.00 [-0.11, 0.03] |

Parameter values are shown as median [IQR]. Difference of parameter values between the two conditions was tested by a sign test. $**p<0.01$, $***p<0.001$.

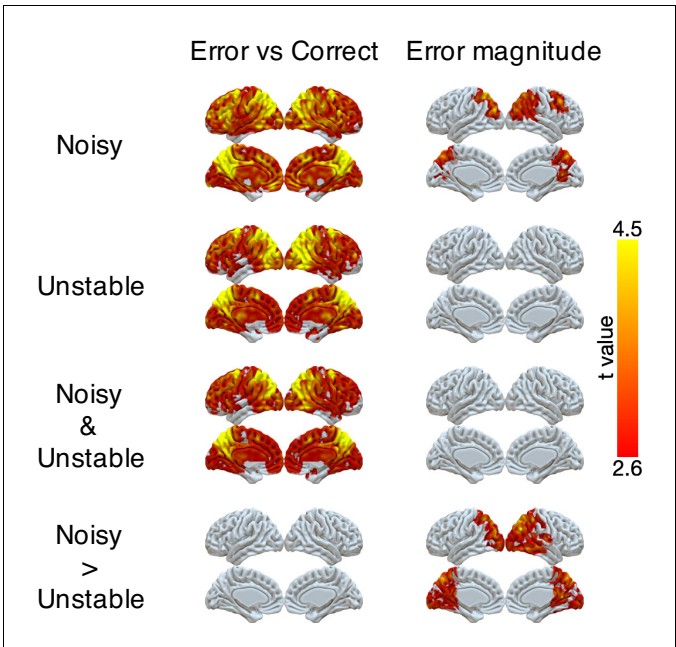

**Figure 3.** Representations of error and error magnitude. For error versus correct analyses, multi-voxel neural patterns were used to classify whether the response on the current trial was correct or an error. For error magnitude analyses, multi-voxel neural patterns were used to classify different error magnitudes (1, 2, 3+) conditional on the current trial being an error. Accuracies were calculated and compared with the baseline accuracy within each subject and then tested at the group level. The representation of current error magnitude is stronger in parietal cortex in the noisy condition than the unstable condition. The cluster-forming threshold was an uncorrected voxel p<0.01 (*t* = 2.6), with cluster mass corrected for multiple comparisons using non-parametric permutation tests.

The online version of this article includes the following figure supplement(s) for figure 3:

**Figure supplement 1.** Univariate representations of error and error magnitude.

analyses, we found that univariate activity in PPC also varied in a task-dependent way, responding more strongly to error magnitude in the noisy than the unstable condition (*Figure 3—figure supplement 1*).

We also found robust, task-dependent representations of past errors in PPC. These representations also were stronger in the noisy than the unstable condition, particularly on trials for which past errors had the strongest influence on behavior. Specifically, we could classify correct versus error on the previous trial in PPC for both task conditions (*Figure 4*). This classification of past errors depended on the outcome of the current trial. We separated trials according to whether the current feedback was correct or an error, or whether the error magnitude provided ambiguous (error magnitudes of 1 or 2) or unambiguous (error magnitudes of 0 or 3+) feedback in the noisy condition (*Figure 4*). We found reliable classifications of past errors in the lateral and medial parietal cortex in both conditions for correct trials and trials with error magnitudes of 0 or 3+. Moreover, these representations depended on the current condition, and, consistent with behavioral effects of error history, were stronger for error trials and trials with error magnitudes of 1 or two in the noisy than in the unstable condition (*Table 3*). These task-dependent signals for past errors were not clearly present in univariate activity (*Figure 4—figure supplement 1*). An additional conjunction analysis across MVPA results showed that PPC uniquely encoded task-dependent error signals for both error magnitude of the current trials and past errors when the current trial's error magnitude was 1 or 2 (*Table 3*).

## Neural prediction of subsequent changes in behavior

Although PPC responds to errors in a task-dependent manner that could be used for determining behavioral updates, we did not find that activity in this region was predictive of the participants'

**Table 3.** Summary of fMRI results: error magnitude and past error.

| Cluster index | #Voxels | Region | Peak t | Peak x | Peak y | Peak z |
|---|---|---|---|---|---|---|
| Error magnitude: noisy versus unstable | | | | | | |
| 1 | 21032 | R precuneus | 5.22 | 16 | −56 | 12 |
| | | R angular gyrus | 5.17 | 44 | −70 | 32 |
| | | L precuneus | 5.08 | −18 | −58 | 20 |
| | | Occipital pole | 5.07 | 2 | −98 | -2 |
| | | L superior parietal lobule | 4.91 | −10 | −66 | 48 |
| | | R occipital cortex | 4.69 | 26 | −76 | 18 |
| | | L occipital cortex | 4.54 | −38 | −86 | 26 |
| | | R superior parietal lobule | 4.44 | 44 | −44 | 54 |
| | | Posterior cingulate cortex | 4.43 | 2 | −46 | 20 |
| Past error on current error magnitude of 1 or 2: noisy versus unstable | | | | | | |
| 1 | 1881 | Posterior cingulate cortex | 4.79 | 12 | −24 | 52 |
| | | R Superior parietal lobule | 4.04 | 32 | −38 | 54 |
| | | R Precuneus | 3.58 | 6 | −54 | 70 |
| | | L superior parietal lobule | 3.54 | −16 | −54 | 62 |
| Conjunction: Error magnitude and Past error on current error magnitude of 1 or 2 | | | | | | |
| 1 | 304 | R superior parietal lobule | 3.41 | 38 | −40 | 52 |
| 2 | 103 | R Precuneus | 3.02 | 2 | −58 | 70 |
| 3 | 81 | L superior parietal lobule | 3.23 | −18 | −56 | 72 |

future behavior. Instead, we found such predictive activity more anteriorly in the frontal lobe. Specifically, we investigated whether multi-voxel neural patterns could predict participants' target switches on the subsequent trial. We focused on the trials with small error magnitudes (1 or 2) in the noisy condition, because these were the only trial types that participants consistently exhibited an intermediate probability of switching (20–80%, *Figure 2*). We found that activity patterns in large cluster encompassing motor cortex, OFC, ACC, DMFC, and DLPFC could predict subsequent stay/switch decisions (*Figure 5*, *Table 4*). We also evaluated this result with different approaches to cluster formation that were more or less spatially specific (*Figure 5—figure supplement 1*). We did not find any regions where univariate activity reliably predicted participants' subsequent behavior (*Figure 5—figure supplement 2*).

## Discussion

We identified task-dependent neural representations of errors in humans performing dynamic learning tasks. Participants were required to learn in two different dynamic environments. In the unstable condition (high-hazard rate and low noise), errors unambiguously indicated a change in the state of the environment, and participants reliably updated their behavior in response to errors. In contrast, in the noisy condition (low-hazard rate and high noise), small errors were ambiguous, and participants used both the current error magnitude and recent error history to distinguish between those errors that likely signal change-points and those likely arising from environmental noise. Using MVPA, we showed complementary roles of PPC and prefrontal regions (including motor cortex, OFC, ACC, DMFC and DLPFC) in the outcome-monitoring and action-selection processes underlying these flexible, task-dependent behavioral responses to errors. Neural patterns in PPC encoded the magnitude of errors and past errors, more strongly in the noisy than the unstable condition. These task-dependent neural responses to errors in PPC were not reliably linked to subsequent changes in behavior. In contrast, neural patterns in prefrontal regions could predict subsequent changes in behavior (whether participants switch their choice on the next trial or not) in response to ambiguous errors in the noisy condition.

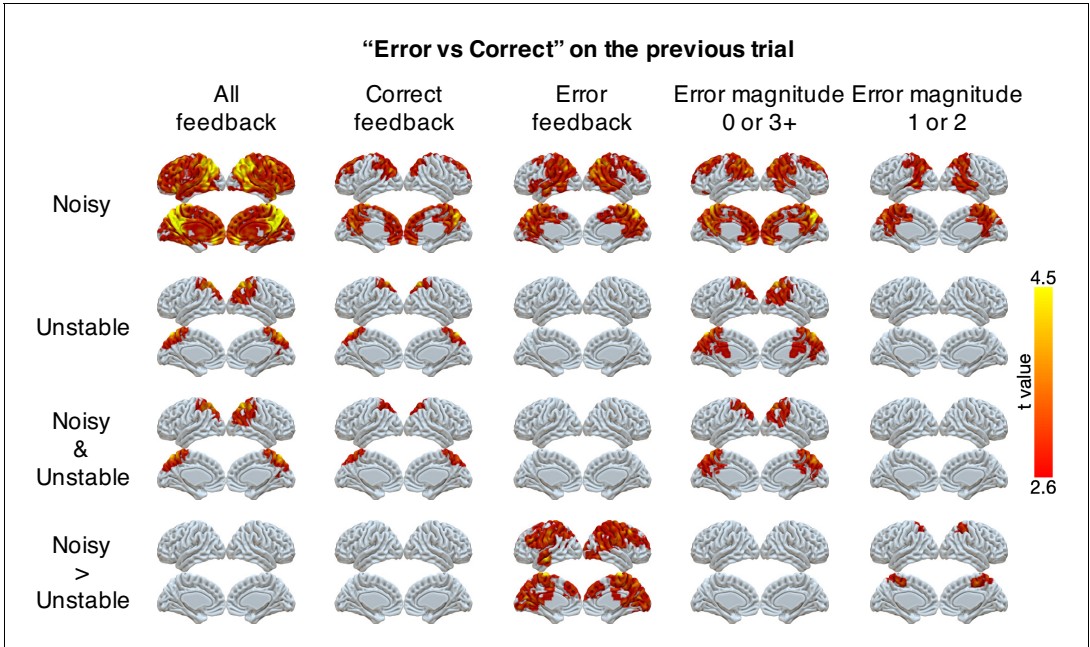

**Figure 4.** Representations of errors on the previous trial conditional on different types of current trials (columns). Multi-voxel neural patterns were used to classify correct responses versus errors on the previous trial. This analysis was repeated for different types of current trials: all feedback, correct feedback, error feedback, error magnitude of 0 or 3+, and error magnitude of 1 or 2. The representation of past errors is stronger in parietal cortex in the noisy condition than the unstable condition when the current trial is an error or the current error magnitude is 1 or 2. The cluster-forming threshold was an uncorrected voxel p<0.01 ($t$ = 2.6), with cluster mass corrected for multiple comparisons using non-parametric permutation tests.

The online version of this article includes the following figure supplement(s) for figure 4:

**Figure supplement 1.** Univariate representations of error on the previous trial conditional on different types of current trials (columns).

## Task-dependent behavior adaptation

Consistent with previous studies of ours and others (*d'Acremont and Bossaerts, 2016*; *McGuire et al., 2014*; *Nassar et al., 2019a*; *Nassar et al., 2012*; *Nassar et al., 2010*; *O'Reilly et al., 2013*; *Purcell and Kiani, 2016*), human participants adapted their response to errors differently in different environments. In the unstable condition, participants almost always switched their choice after errors and quickly learned the new state after change-points. In contrast, in the noisy condition, participants ignored many errors and only slowly learned the new state after change-points. In this condition, participants had to distinguish true change-points from environmental noise, and they used error magnitude and recent error history as a cue for whether the state had recently changed or not. These flexible and task-dependent responses to errors could be accounted for by a reduced Bayesian model (*McGuire et al., 2014*; *Nassar et al., 2012*; *Nassar et al., 2010*). This model assumes that participants use approximately optimal inference processes but can have subjective estimates of environmental parameters (hazard rate, noise) that depart from their true values.

## Neural representation of change-point probability and relative uncertainty

In the reduced Bayesian model, beliefs and behavior are updated dynamically according to two key internal quantities, CPP and RU. Replicating our previous work (*McGuire et al., 2014*), we identified neural activity correlated with both CPP and RU in PPC and DLPFC. This replication shows the robustness of these neural representations of CPP and RU across experimental designs that differ dramatically in their visual stimuli and motor demands, yet share the need to learn in dynamic environments with similar statistics. We extended those findings to show that some brain regions that encode both CPP, which in the model accounts for task-dependent behavioral responses to error magnitude, and RU, which in the model accounts for task-dependent behavioral responses to recent

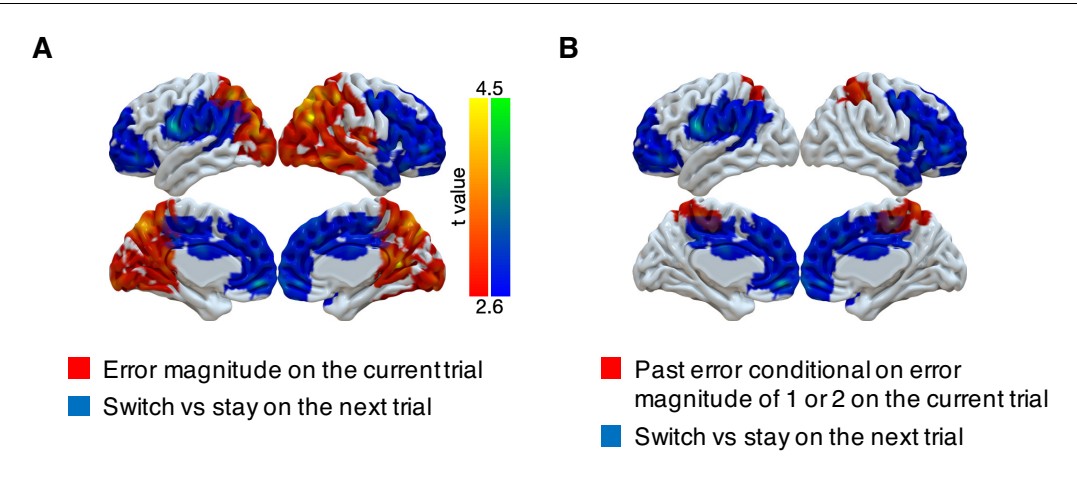

**Figure 5.** Representations of subsequent behavioral choices (switch versus stay) after ambiguous small errors in the noisy condition. (**A**) Overlap of results for switch versus stay on the next trial and error magnitude on the current trial. Multi-voxel neural patterns were used to classify whether participants switch their choice to another target or stay on the same target on the next trial. We focused on the most ambiguous errors (error magnitude of 1 or two in the noisy condition). Above-chance classification performance was found in a large cluster encompassing the frontal lobe. The cluster-forming threshold was an uncorrected voxel p<0.01 (t = 2.6), with cluster mass corrected for multiple comparisons using non-parametric permutation tests. (**B**) Overlap of results for switch versus stay on the next trial and past error conditional on error magnitude of 1 or two on the current trial.

The online version of this article includes the following figure supplement(s) for figure 5:

**Figure supplement 1.** Representations of subsequent behavioral choices (switch versus stay) thresholded via threshold-free cluster enhancement (TFCE) or with a cluster-forming threshold of p<0.001.

**Figure supplement 2.** Univariate GLM for switch versus stay on small error trials (magnitudes of 1 or 2) in the noisy condition.

error history, also encode errors in a task-dependent manner or predict subsequent behavioral updates.

**Table 4.** Summary of fMRI results: behavior change.

| Cluster index | #Voxels | Region | Peak t | Peak x | Peak y | Peak z |
|---|---|---|---|---|---|---|
| Switch versus stay on error magnitude of 1 or two in the noisy condition | | | | | | |
| 1 | 12042 | Middle cingulate cortex | 4.35 | 14 | -8 | 30 |
| | | R insula | 4.33 | 38 | 4 | 2 |
| | | Medial orbitofrontal cortex | 4.24 | -4 | 50 | −10 |
| | | R frontal pole | 4.11 | 40 | 46 | 0 |
| | | R inferior frontal gyrus | 4.11 | 48 | 26 | 10 |
| | | L frontal pole | 4.01 | −24 | 52 | -2 |
| | | Dorsomedial frontal cortex | 3.96 | 0 | 26 | 34 |
| | | Posterior cingulate cortex | 3.93 | 2 | −28 | 50 |
| | | R primary motor cortex | 3.91 | 48 | -6 | 50 |
| | | Anterior cingulate cortex | 3.51 | 0 | 48 | 20 |
| 2 | 3134 | L premotor cortex | 4.43 | −62 | 2 | 24 |
| | | L superior temporal gyrus | 4.28 | −50 | −32 | 12 |
| | | L inferior frontal junction | 3.72 | −38 | 4 | 28 |
| | | L postcentral gyrus | 3.61 | −50 | −26 | 44 |

## Task-dependent neural representation of errors

Advancing beyond previous work, we identified task-dependent encoding of errors in neural activity in the PPC. Mirroring the task dependence of behavior, the multivariate neural pattern in PPC encoded current error magnitude more strongly in the noisy condition than in the unstable condition and encoded past errors more strongly on trials that provided ambiguous feedback in the noisy condition. These same regions of PPC have been shown previously to represent errors, error magnitudes, surprise and salience (*Fischer and Ullsperger, 2013*; *Gläscher et al., 2010*; *McGuire et al., 2014*; *Nassar et al., 2019a*; *Nassar et al., 2019b*; *O'Reilly et al., 2013*; *Payzan-LeNestour et al., 2013*). In addition, these regions have been shown to integrate recent outcome or stimulus history in human fMRI studies (*FitzGerald et al., 2015*; *Furl and Averbeck, 2011*) and in animal single neuron recording studies (*Akrami et al., 2018*; *Brody and Hanks, 2016*; *Hanks et al., 2015*; *Hayden et al., 2008*; *Hwang et al., 2017*). Our results extend on these past findings by demonstrating that the neural encoding of error magnitude and error history in PPC is modulated across different conditions in precisely the manner that could drive adaptive behavior.

These whole-brain fMRI results complement our previous results recording from single neurons in ACC and PCC in the same task (*Li et al., 2019*). In that study, we identified single neurons in both ACC and PCC that encoded information relevant to interpreting errors, such as the magnitude of the error or the current condition. However, we did not find any neurons that combined this information in a manner that could drive adaptive behavioral adjustments. Our whole-brain fMRI results suggest that PPC would be a good place to look for task-dependent error representations in single neurons, including a region of medial parietal cortex slightly dorsal to the PCC area we recorded from previously.

## Neural representations of behavioral updating

Also advancing beyond previous work, we identified neural activity predictive of behavioral updates across the frontal cortex, including DLPFC. In the noisy condition, small errors provided ambiguous feedback that could reflect either a change in state or environmental noise. Accordingly, after small errors in the noisy condition, participants exhibited variability across trials in whether they switched from their current choice on the subsequent trial or not. In these ambiguous situations, the multivariate neural pattern in large cluster in frontal cortex, including motor cortex, OFC, ACC, DMFC and DLPFC, predicted whether people switched or stayed on the subsequent trial. These results extend previous findings that the multivariate pattern in frontal cortex, particularly ACC and medial PFC, can decode subsequent switching versus staying in a reversal learning task (*Hampton and O'doherty, 2007*). These results suggest a dissociation between PPC regions that monitor error information in a task-dependent manner and frontal regions that may use this information to update beliefs and select subsequent actions.

This ability to decode subsequent choices might arise from different kinds of representations in different areas of frontal cortex. Whereas motor and premotor regions may reflect the change in action plans, other frontal regions might reflect changes in abstract representations of belief states. Medial PFC is involved in performance monitoring, distinguishing errors from different sources such as actions and feedback (*Ullsperger et al., 2014*), registering a hierarchy of prediction errors from those due to environmental noise to those due to a change in the environmental state (*Alexander and Brown, 2015*), and interacting with lateral PFC to guide subsequent behavioral adjustments in response to errors (*Alexander and Brown, 2015*). Consistent with this role, activity in DMFC also reflects the extent of belief updating in dynamic environments (*Behrens et al., 2007*; *Hampton et al., 2006*; *McGuire et al., 2014*; *O'Reilly et al., 2013*). OFC and DMFC encode the identity of the current latent state in a mental model of the task environment and neural representations in these regions changes as the state changes (*Chan et al., 2016*; *Hunt et al., 2018*; *Karlsson et al., 2012*; *Nassar et al., 2019b*; *Schuck et al., 2016*; *Wilson et al., 2014*). Activity in inferior frontal junction reflects the updating of task representations (*Brass and von Cramon, 2004*; *Derrfuss et al., 2005*). Neural activity in frontopolar cortex (*Daw et al., 2006*) and DMFC (*Blanchard and Gershman, 2018*; *Kolling et al., 2012*; *Kolling et al., 2016*; *Muller et al., 2019*) increases during exploratory choices, which occur more frequently during periods of uncertainty about the most beneficial option. In a recent study, we identified distinct representations of latent states, uncertainty, and behavioral policy in distinct areas of frontal cortex during learning in a

dynamic environment (*Nassar et al., 2019b*). Our results extend these past findings and demonstrate the role of these frontal regions in adjusting behavior in response to ambiguous errors.

## Caveats

A few caveats should be considered when interpreting our results. First, we had relatively small number of participants in this study (n = 16). Although we control the false-positive rates through permutation tests that have been validated empirically (*Eklund et al., 2016*), it is possible that we lacked the statistical power to detect some effects, and so null results should be interpreted with caution. Second, in this study, we created two qualitatively different task conditions by manipulating both the noise levels and hazard rates. Thus, we cannot attribute any behavioral or neural differences across conditions specifically to changes in either noise levels or hazard rates alone, but rather to how the combinations of these two variables affect the interpretation and use of small errors. Future studies can manipulate hazard rate and noise independently to examine their independent contributions to adaptive learning.

## Conclusion

People adapt their behavior in response to errors in a task-dependent manner, distinguishing between errors that indicate change-points in the environment versus noise. Here we used MVPA to identify two distinct kinds of neural signals contributing to these adaptive behavioral adjustments. In PPC, neural patterns encoded error information in a task-dependent manner, depending on error magnitude and past errors only under conditions where these were informative of the source of error. In contrast, activity in frontal cortex could predict subsequent choices that could be based on this information. These findings suggest a broad distinction between outcome monitoring in parietal regions and action selection in frontal regions when learning in dynamic and uncertain environments.

# Materials and methods

## Participants

All procedures were approved by University of Pennsylvania Internal Review Board. We analyzed data from sixteen participants (nine females, seven males, mean age = 23.5, SD = 4.3, range = 18–33 years) recruited for the current study. One additional participant was excluded from analyses because of large head movements during MRI scanning (>10% of timepoint-to-timepoint displacements were >0.5 mm). All participants provided informed consent before the experiment. Participants received a participation fee of $15, as well as extra incentives based on their performance (mean = $15.09, SD = $2.26, range = $8.5–17.5).

## Task

Participants performed a predictive-inference task during MRI scanning. On each trial, participants saw a noisy observation sampled from an unobserved state. The participants' goal was to predict the location of the noisy observation. To perform this task well, however, they should infer the location of the current state.

In this task (*Li et al., 2019*), there were 10 targets aligned in a circle on the screen (*Figure 1A*). At the start of each trial, participants had to fixate a central cross for 0.5 s to initialize the trial. After the cross disappeared, participants could choose one of 10 targets (red) by looking at it within 1.5 s and keeping fixation on the chosen target for 0.3 s. Then, an outcome would be shown for 1 s. During the outcome phase, a green dot indicated the chosen target. A purple or cyan target indicated the rewarded target, with color denoting 10 or 20 points of reward value, respectively. At the end of experiment, every 75 points were converted to $0.25 as participants' extra incentives.

Participants performed this task in two dynamic conditions separated into two different runs: a high-noise/low-hazard ('noisy') condition and an low-noise/high-hazard ('unstable') condition. In the noisy condition, the rewarded target could be one of five targets, given the underlying state (*Figure 1B*). The rewarded target probabilities for the relative locations ([−2,–1, 0, 1, 2]) of the current state were [0.05, 0.15, 0.6, 0.15, 0.05]. Thus, the location of the current state was most likely rewarded, but nearby targets could also be rewarded. Occasionally, the state would change its location with a hazard rate of 0.02 (*Figure 1C*). When a change-point happens, the new state would be

selected among the ten targets based on a uniform distribution. In the unstable condition, there was no noise (*Figure 1D*). That is, the location of the state would be always rewarded. However, the state was unstable, as the hazard rate in this condition was 0.35 (*Figure 1E*). There were 300 trials in each run.

## Behavior analysis

We investigated how participants used error feedback flexibly across different conditions. Before the behavioral analysis, we removed two different kinds of trials. First, we removed trials in which participants did not make a choice within the time limit (Unstable: median number of trials = 10.5, range = 1–83; Noisy: median = 10, range = 2–88). Second, we also removed trials in which the location of the chosen target was not on the shortest distance between the previously chosen and previously rewarded targets (Unstable: median = 3, range = 0–24; Noisy: median = 17, range = 5–37). All of the belief-updating models we tested predict that participants' choice should be along the shortest distance between the previously chosen target and the previously rewarded target. That is, participants should update in a clockwise direction, if the shortest distance to rewarded target was clockwise of the chosen target. Otherwise, they should update in a counterclockwise direction. We removed trials where participants' update was in the opposite direction of the rewarded target (which would correspond to a learning rate <0) and trials where participants' update was beyond the location of the rewarded target (which would correspond to a learning rate >1), as this behavior cannot be captured by any of the belief-updating models we tested. Further, this behavior might suggest that participants had lost track of the most recently chosen or rewarded targets. The analysis codes were written in MATLAB and are available at Github (*Kao, 2020*; https://github.com/chan-ghaokao/mvpa_changepoint_fmri).

First, we investigated how fast participants learned the location of the current state. For each condition and participant, we binned trials from trial 0 to trial 20 after change-points. Then, we calculated the probability of choosing the location of the current state for each bin.

Second, we examined how different magnitudes of errors lead to shifts in behavior. For each condition and participant, we binned trials based on the current error magnitude (from 0 to 5). Then, for each bin, we calculated the probability that participants switch their choice to another target on the subsequent trial. We hypothesized participants would have a lower probability of switching after small error magnitudes (1 or 2) in the noisy condition than in the unstable condition since such errors could be due to environment noise in the noisy condition but would signal a state change in the unstable condition.

Third, we further investigated how error history influenced participants' behavioral shifts. Similarly, we binned trials based on the current error magnitude and the error history of the last three trials. Here, we used four bins of error magnitudes (0, 1, 2, 3+). Based on the outcome of correct or error on the last three trials, there were 8 types of error history. For each error magnitude, we calculated the probability of switching for each type of error history. We hypothesized that participants in the noisy condition would tend to switch their choice after small errors more if they had made more errors recently. To test this hypothesis, we ordered the 8 types of error history based on the number of recent errors and calculated the slope of probability of switching against the order of error history. A negative slope means that participants tend to switch as they receive more recent errors.

## Behavior modeling

We fit several different computational models to participants' choices to evaluate which ones could best account for their behavior in the task.

## Reduced Bayesian (RB) model

Previous studies have shown that a reduced Bayesian model, which approximates the full Bayesian ideal observer, could account well for participants' behavior in dynamic environments similar to the current task (*McGuire et al., 2014*; *Nassar et al., 2012*; *Nassar et al., 2010*). In this model, belief is updated by a delta rule:

$$\delta_t = x_t - B_t \tag{1}$$

$$B_{t+1} = B_t + \alpha_t \delta_t \qquad (2)$$

where $B_t$ is the current belief and $x_t$ is the current observation. The new belief ($B_{t+1}$) is formed by updating the old belief according to the prediction error ($x_t - B_t$) and a learning rate ($\alpha_t$). The learning rate controls how much a participant revises their belief based on the prediction error. In this model, the learning rate is adjusted on a trial-by-trial basis according to:

$$\alpha_t = \Omega_t + (1 - \Omega_t)\tau_t \qquad (3)$$

where $\Omega_t$ is the change-point probability and $\tau_t$ is the relative uncertainty. That is, $\alpha_t$ is high as either $\Omega_t$ or $\tau_t$ is high. The change-point probability is the relative likelihood that the new observation represents a change-point as opposed to a sample from the currently inferred state (**Nassar et al., 2010**):

$$\Omega_t = \frac{U(x_t|1,\ 10)H}{U(x_t|1,\ 10)H + f_p(x_t|\gamma_t, B_t)(1-H)} \qquad (4)$$

where $H$ is the hazard rate, $U(x_t|1,\ 10)$ is the probability of outcome derived from a uniform distribution, and $f_p(x_t|\gamma_t, B_t)$ is the probability of outcome derived from the current predictive distribution. That is, $U(x_t|1,\ 10)$ reflects the probability of outcome when a change-point has occurred while $f_p(x_t|\gamma_t, B_t)$ reflects the probability of outcome when the state has not changed.

The predictive distribution is an integration of the state distribution and the noise distribution:

$$f_p(X|\gamma_t, B_t) = C \times P(X|B_t)^{\gamma_t} \times P(X|B_t) \qquad (5)$$

where $X$ is a random variable determining the locations of target, $P(X|B_t)$ is the noise distribution in the current condition, $P(X|B_t)^{\gamma_t}$ is the state distribution, $\gamma_t$ is the expected run length after the change-point, and $C$ is a normalizing constant to make the sum of probabilities in the predictive distribution equal one. Thus, the uncertainty of this predictive distribution comes from two sources: the uncertainty of the state distribution ($\sigma_s^2$) and the uncertainty of the noise distribution ($\sigma_N^2$). The uncertainty of the state distribution would decrease as the expected run length increases.

The expected run length reflects the expected number of trials that a state remains stable, and thus is updated on each trial based on the change-point probability (**Nassar et al., 2010**):

$$\gamma_{t+1} = (\gamma_t + 1)(1 - \Omega_t) + \Omega_t \qquad (6)$$

where the expected run length is a weighted average conditional on the change-point probability. If no change-point occurs (i.e. change-point probability is low), the expected run length would increase, leading the uncertainty of the state distribution to decrease. That is, as more observations from the current state are received, participants are more certain about the location of the current state. However, if the change-point probability is high, which signals a likely change in the state, the expected run length would be reset to 1. Thus, the uncertainty of the state distribution becomes large. Participants are more uncertain about the current state after a change-point.

The other factor influencing the learning rate is the relative uncertainty, which is the uncertainty regarding the current state relative to the irreducible uncertainty or noise (**McGuire et al., 2014**; **Nassar et al., 2012**):

$$\tau_{t+1} = \frac{\Omega_t \sigma_N^2 + (1 - \Omega_t)\sigma_s^2 + \Omega_t(1 - \Omega_t)[\delta_t(1 - \tau_t)]^2}{\Omega_t \sigma_N^2 + (1 - \Omega_t)\sigma_s^2 + \Omega_t(1 - \Omega_t)[\delta_t(1 - \tau_t)]^2 + \sigma_N^2} \qquad (7)$$

The three terms in the numerator contribute to the uncertainty about the current state. The first term reflects the uncertainty conditional on the change-point distribution; the second term reflects the uncertainty conditional on the non-change-point distribution; and the third term reflects the uncertainty due to the difference between the two distributions. The denominator shows the total variance which is the summation of the uncertainty about the current state and the noise. As more precise observations are received in a given state, this relative uncertainty would decrease.

To fit the reduced Bayesian model to behavior, we assumed that participants can depart from the ideal observer by having subjective estimates of the key environmental variables, hazard rate and

noise, that may differ from the true value of these variables. During model fitting, the subjective noise distribution was estimated with the von Mises distribution, which is a circular Gaussian distribution:

$$P(x_t|B_t,K) = \frac{e^{K\cos(x_t - B_t)}}{\sum\limits_{i=1}^{10} e^{K\cos(x_i - B_t)}}$$ (8)

where $B_t$ is the location of the current belief, $x_i$ is the location of target, and $K$ controls the uncertainty of this distribution. When $K$ is 0, this is a uniform distribution. As $K$ increases, the uncertainty decreases. The denominator is used as a normalization term to make sure the sum of all the probabilities equals one. Thus, there are two free parameters in this model: hazard rate ($H$, in *Equation 4*) and noise level ($K$, in *Equation 8*). The range of hazard rate is between 0 and 1 and the noise level is greater than or equal to zero.

### Fixed learning rate (fixedLR) model

We also consider an alternative model in which participants used a fixed learning rate in each of the two dynamic conditions. That is, the learning rate is the same over all trials in a condition. This model has one free parameter, the fixed learning rate ($\alpha_{fixed}$), for each condition (*Equation 2*). The fixed learning rate is between 0 and 1.

### Hybrid of RB model and fixedLR model

Furthermore, we consider a hybrid model, in which the learning rate on each trial is a mixture of the learning rates from the RB model and the fixedLR model:

$$\alpha_t = w\alpha_{RB} + (1 - w)\alpha_{fixed}$$ (9)

where $\alpha_{RB}$ is the learning rate from the RB model and is varied trial-by-trial according to $\Omega_t$ and $\tau_t$, $\alpha_{fixed}$ is the learning rate from the fixedLR model and $w$ reflects the weight to integrate these two learning rates. In this model, there are four free parameters: hazard rate, noise level, fixed learning rate and weight. The weight is between 0 and 1.

### Hybrid of RB model and P~stay~

### Hybrid of RB model and $P_{stay}$

Finally, we consider a hybrid model, which combines the RB model with a fixed tendency to stay on the current target regardless of the current observation. Such a fixed tendency to stay was observed in monkeys in our previous study (*Li et al., 2019*). Here the belief is updated by:

$$B_{t+1} = B_t + \left[ (1 - P_{stay}) \times \alpha_t(X_t - B_t) + P_{stay} \times 0 \right]$$ (10)

where $P_{stay}$ is the probability that participants stay on the current target. This model has three free parameters: hazard rate, noise level and the probability of stay. The probability of stay is between 0 and 1.

### Model fitting and comparison

Each model was fitted to data from each participant and within each condition separately. Optimal parameters were estimated by minimizing the mean of the squared error (MSE) between a participant's prediction and the model prediction.

$$MSE = \frac{\sum\limits_{t=1}^{n} \left( B_t - B_t \right)^2}{n}$$ (11)

where $t$ is the trial, $n$ is the total number of included trials, $B_t$ is a participant's prediction on trial $t$, and $B_t$ is the model prediction on trial $t$.

Because each model used a different number of parameters and each participant had a different number of included trials, we used Bayesian Information Criterion (BIC) to compare the performance of different models:

$$BIC = n \ln(MSE) + k \ln(n) \tag{12}$$

where $n$ is the number of included trials and $k$ is the number of free parameters in a model. A model with lower BIC performs better.

## MRI data acquisition and preprocessing

We acquired MRI data on a 3T Siemens Prisma with a 64-channel head coil. Before the task, we acquired a T1-weighted MPRAGE structural image ($0.9375 \times 0.9375 \times 1$ mm voxels, $192 \times 256$ matrix, 160 axial slices, TI = 1,100 ms, TR = 1,810 ms, TE = 3.45 ms, flip angle = 9°). During each run of the task, we acquired functional data using a multiband gradient echo-planar imaging (EPI) sequence ($1.9592 \times 1.9592 \times 2$ mm voxels, $98 \times 98$ matrix, 72 axial slices tilted 30° from the AC-PC plane, TR = 1,500 ms, TE = 30 ms, flip angle = 45°, multiband factor = 4). The scanning time (mean = 24.14 min, SD = 1.47, range = 21.85–30.00) for each run was dependent on the participants' pace. After the task, fieldmap images (TR = 1,270 ms, TE = 5 ms and 7.46 ms, flip angle = 60°) were acquired.

Data were preprocessed using FMRIB's Software Library (FSL) (*Jenkinson et al., 2012*; *Smith et al., 2004*). Functional data were motion corrected using MCFLIRT (*Jenkinson et al., 2002*), high-pass filtered with a Gaussian-weighted least square straight line fitting of $\sigma = 50\ s$, undistorted and warped to MNI space. To map the data to MNI space, boundary-based registration was applied to align the functional data to the structural image (*Greve and Fischl, 2009*) and fieldmap-based geometric undistortion was also applied. In addition, the structural image was normalized to the MNI space (FLIRT). Then, these two transformations were applied to the functional data.

## fMRI analysis: univariate activity correlated with CPP and RU

Using similar procedures to our previous study (*McGuire et al., 2014*), we examined the effects of CPP and RU on univariate activity. Both the current study and the previous study investigate the computational process and neural mechanisms during learning in dynamic environments. The underlying task structures (which involved noisy observations and sudden change-points) are similar between the two studies, but the two studies used very different visual stimuli and motor demands. We specifically focused on the noisy condition in the current study because it was more similar to the underlying structure, in terms of noisy observations and hazard rate of change-points, to our previous study.

We investigated the factors of CPP, RU, reward values and residual updates. The trial-by-trial CPP and RU were either estimated from the RB model with subjective estimates of hazard rate and noise (as this was the best-fitting model in the current study, analyses presented in *Figure 2—figure supplement 3*) or from the RB model with true estimates of hazard rate and noise (as this corresponds to how correlates of CPP and RU were identified in our previous study, analyses presented in *Figure 2—figure supplement 4*). The residual update reflects the difference between the participants' update and the predicted update, and is estimated from a behavioral regression model in a similar manner as our previous study:

$$Update_t = \beta_0 + \beta_1 \delta_t + \beta_2 \delta_t \Omega_t + \beta_3 \delta_t (1 - \Omega_t) \tau_t + \beta_4 \delta_t Reward + \varepsilon \tag{13}$$

where $Update_t$ is the difference between $B_{t+1}$ and $B_t$, $\delta_t$ is the error magnitude, both $\Omega_t$ and $\tau_t$ were derived from the RB model, and the reward value indicated whether a correct response earned a large or a small value on that trial.

Then, a general linear model using these four factors was implemented on the neural data. Here we further smoothed the preprocessed fMRI data with a 6 mm FWHM Gaussian kernel. We included several trial-by-trial regressors of interest in the GLM: onsets of outcome, CPP, RU, reward value, and residual update. Six motion parameters were also included as confounds. To control false-positive rates (*Eklund et al., 2016*), statistical testing was implemented through one-sample cluster-mass permutation tests with 5000 iterations. The cluster-forming threshold was uncorrected voxel p<0.01. Statistical testing was then based on the corrected cluster *p* value. For the conjunction analyses, we used the same procedure as the previous study (*McGuire et al., 2014*). We kept regions that passed the corrected threshold and showed the same sign of effects. For these conjunction tests, we only kept regions that have at least 10 contiguous voxels.

Because the number of participants was fewer in this study (n = 16) than in the previous study (n = 32), we might have lower power to detect effects in the whole-brain analyses. Thus, we also implemented ROI analyses. We selected seven ROIs that showed the conjunction effects of CPP, RU and reward value in the previous study (*McGuire et al., 2014*) and tested the effects of CPP and RU in these ROIs.

We found previously that for a similar task, the presence or absence of reward on a given trial influenced both belief-updating behavior and some aspects of its neural representation (*McGuire et al., 2014*). To further examine those effects, here we included two different earnable values (10 versus 20 points). However, we did not find any significant effects of earnable values on either belief updating ($\beta_4$ in *Equation 13* was not significantly different than zero) or neural activity (for the contrast of high versus low earnable value). We therefore do not further consider the effects of this manipulation in the current report. We speculate that this lack of an effect contrasts from our earlier finding because here we used high versus low earnable values, whereas in that study we used the presence versus absence of earnable value.

## fMRI analysis: multi-voxel pattern analysis (MVPA)

We implemented MVPA to understand the neural representation of error signals and subsequent choices. Our analyses focus on the multi-voxel pattern when participants received an outcome. Before implementing MVPA, we estimated trial-by-trial beta values using the unsmoothed prepro-cessed fMRI data. We used the general linear model (GLM) to estimate the beta weights for each trial (*Mumford et al., 2012*). In each GLM, the first regressor is the trial of interest and the second combines the rest of trials in the same condition. These two regressors were then convolved with a gamma hemodynamic response function. In addition, six motion parameters were included as control regressors. We repeated this process (one GLM per trial) to estimate trial-by-trial beta values for all the trials in the two conditions. We then used these beta values as observations for MVPA. A whole-brain searchlight was implemented (*Kriegeskorte et al., 2006*). In each searchlight, a sphere with the diameter of 5 voxels (10 mm) was formed, and the pattern of activity across the voxels within the sphere were used to run MVPA.

A support vector machine (SVM) with a linear kernel was used to decode different error signals and choices in our whole-brain searchlight analysis. We implemented SVM through the LIBSVM tool-box (*Chang and Lin, 2011*). To avoid overfitting, we used 3-fold cross-validation, with one fold used as testing data and the other two as training data. Training data were used to train the classifier and then this classifier was used on testing data to examine the classification accuracy. In linear SVM, a free parameter *c* regularizes the trade-off between decreasing training error and increasing generali-zation. Thus, during the training of classifier, the training data were further split into 3-folds to select the optimal value of the parameter *c* through cross-validation. We pick the optimal value for *c* from [0.001, 0.01, 0.1, 1, 10, 100, 1000] and this optimal parameter should maximize the cross-validation accuracy. Then, we used the optimal parameter *c* to train the model again based on the entire train-ing data and calculated the classification accuracy on the testing data. We repeated this procedure with each of the three folds held out as testing data and calculated the average of the classification accuracy. To minimize the influence of different number of trials for each category on the classifica-tion accuracy, we used balanced accuracy. For balanced accuracy, we first calculated the classifica-tion accuracy within each category, and then averaged the accuracies across all categories. The baseline balanced accuracy was also validated via permutations with 5000 iterations. For each per-mutation, each trial was randomly assigned one category with a probability proportional to the num-ber of trials in that category among all the trials. We then used the average of balanced accuracy across these iterations as the baseline accuracy. The baseline accuracy for two categories was 50% and for three categories was 33%.

We first examined how the multi-voxel neural pattern on the current trial could discriminate cor-rect versus error on the current trial or error magnitudes on error trials. For the analysis of error mag-nitudes, we split trials into three bins of error magnitude: 1, 2, and 3+.

We next examined how the multi-voxel neural pattern on the current trial could discriminate whether the previous trial was an error or not. We also investigated how the classification of past errors differs conditional on the type of the current trial. We classified trial *t-1* as correct or error sep-arately for four different types of current trials: correct trials, error trials, trials with error magnitudes of 0 or 3+ and trials with error magnitudes of 1 or 2. We differentiated between trials with error

magnitudes of 0 or 3+ and trials with error magnitudes of 1 or two because error magnitudes of 0 or 3+ provide unambiguous evidence regarding a change of state in the noisy condition while error magnitudes of 1 or two provide ambiguous evidence about a change of the state in the noisy condition.

Lastly, we examined how the multi-voxel neural pattern on the current trial could classify the choice on the next trial. In this analysis, we focused only on the trials with error magnitudes of 1 or two in the noisy condition, because only under these conditions were participants similarly likely to switch versus stay. For these trials, we examined whether the multi-voxel pattern on the current trial predicted whether the participant stayed or switched on the next trial.

After obtaining the classification accuracy for each participant, we subtracted the baseline accuracy from the classification accuracy. Before conducting a group-level test, we smoothed these individual accuracy maps with a 6 mm FWHM Gaussian kernel. To control false-positive rates (*Eklund et al., 2016*), statistical testing was implemented through one-sample cluster-mass permutation tests with 5000 iterations. We used uncorrected voxel $p<0.01$ to form a cluster and estimated the corrected cluster $p$ value for each cluster. For comparison, we report our results using other cluster-forming procedures in supplemental analyses. For the conjunction analyses, we used the same procedure described above.

## Acknowledgements

This work was supported by grants from National Institute of Mental Health (R01-MH098899 to JIG and JWK) and National Science Foundation (1533623 to J.I.G. and J.W.K.). We thank Yin Li for valuable comments; M Kathleen Caulfield for fMRI scanning.

## Additional information

### Competing interests

Joshua I Gold: Senior editor, *eLife*. The other authors declare that no competing interests exist.

### Funding

| Funder | Grant reference number | Author |
| --- | --- | --- |
| National Institute of Mental Health | R01-MH098899 | Joshua I Gold<br>Joseph W Kable |
| National Science Foundation | 1533623 | Joshua I Gold<br>Joseph W Kable |

The funders had no role in study design, data collection and interpretation, or the decision to submit the work for publication.

### Author contributions

Chang-Hao Kao, Conceptualization, Data curation, Formal analysis, Visualization, Writing - original draft, Writing - review and editing; Sangil Lee, Conceptualization, Data curation, Writing - review and editing; Joshua I Gold, Joseph W Kable, Conceptualization, Supervision, Funding acquisition, Writing - review and editing

### Author ORCIDs

Chang-Hao Kao (iD) https://orcid.org/0000-0002-2928-302X
Sangil Lee (iD) http://orcid.org/0000-0002-4443-9926
Joshua I Gold (iD) http://orcid.org/0000-0002-6018-0483

### Ethics

Human subjects: All procedures were approved by University of Pennsylvania Internal Review Board. All participants provided informed consent before the experiment. (IRB Protocol # 816727).

Decision letter and Author response
Decision letter https://doi.org/10.7554/eLife.58809.sa1
Author response https://doi.org/10.7554/eLife.58809.sa2

## Additional files

### Supplementary files
• Transparent reporting form

### Data availability

The fMRI dataset has been made available at OpenNeuro. Codes and behavioral dataset are available at Github (https://github.com/changhaokao/mvpa_changepoint_fmri; copy archived at https://github.com/elifesciences-publications/mvpa_changepoint_fmri).

The following dataset was generated:

| Author(s) | Year | Dataset title | Dataset URL | Database and Identifier |
|---|---|---|---|---|
| Kao C-H, Lee S, Gold JI, Kable JW | 2020 | changepoint fmri | https://doi.org/10.18112/openneuro.ds003170.v2.0.0 | OpenNeuro, 10.18112/openneuro.ds003170.v2.0.0 |

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
