## [Decision Letter]

**Acceptance summary:**

This study shows that the human brain encodes errors in a task-dependent manner. Similar objective errors elicit different neural responses depending on their meaning, defined by the statistics of the task. These results provide novel insights into how the human brain supports adaptive behavior.

**Decision letter after peer review:**

Thank you for submitting your article "Distinct neural encoding of context-dependent errors and context-dependent changes in behavior during adaptive learning" for consideration by *eLife*. Your article has been reviewed by three peer reviewers, one of whom is a member of our Board of Reviewing Editors, and the evaluation has been overseen by a Reviewing Editor and Timothy Behrens as the Senior Editor. The reviewers have opted to remain anonymous.

The reviewers have discussed the reviews with one another and the Reviewing Editor has drafted this decision to help you prepare a revised submission.

Summary:

This manuscript reports an fMRI study examining error processing in a predictive inference task. The task involves different conditions/contexts defined by their level of noise and volatility. Using MVPA, the authors identified differences in the neural representations of error magnitude and error history in posterior parietal cortex between these conditions. Moreover, behavioral adjustments could be decoded from activity in dorsomedial frontal, anterior cingulate and orbitofrontal cortices.

All reviewers agreed that this is an interesting study addressing a timely question on adaptive behavior, and that the manuscript will be informative to readers interested in decision making, learning, and cognitive control. They further found the experimental design and the general approach convincing, and the results robust. However, there were several concerns regarding analysis and interpretation that need to be addressed prior to publication.

Essential revisions:

1) Results in Figure 6 are interesting as they show that behavioral adjustments can be predicted from brain activity, replicating the findings by Hampton and O'Doherty, 2007. However, it is not clear that they can be considered "context-dependent". This analysis is based on one condition only, and it is therefore not clear whether or not these signals are context-dependent. The reviewers understand that because there are no ambiguous errors in the unstable context the behavioral adjustment analysis is not possible in this context, and thus, no comparisons across contexts can be performed. The authors can choose to keep these results in the paper, but they should not refer to them as "context-dependent" anywhere in the manuscript, including the title and Abstract.

2) The experimental design uses noise level and volatility to define contexts. This is potentially problematic. First, we do not know whether differences between conditions are driven by noise levels, hazard rate, or both. This needs to be discussed. Second, and more importantly, because the noise level is zero in one condition, one context is deterministic whereas the other is probabilistic. This fundamentally changes the meaning of errors and error magnitude between the contexts, such that there is no meaning to error magnitude in the unstable condition (in the sense that larger errors would be any different from smaller errors). Instead, it is a nominal signal that informs subjects of what the new target is, such that an error of 1 carries the same information as an error of 6. Why would such a signal be encoded in a ordinal or linear fashion as in the high-noise condition, where the magnitude of the error is meaningful? Thus, given that contexts are defined by whether or not error magnitude is meaningful, is it really surprising that the brain responds differently to error magnitude? Would the authors expect to find the same results if context was defined by a variable that does not directly affect the meaning of error magnitude (e.g. learning about target locations in different environments)? This boils down to the question of whether these are context-dependent errors, or just fundamentally different errors. Given these issues, reviewers felt that describing the results as "context-dependent" is slightly misleading. To be clear, reviewers feel that the findings are important and interesting, but they agreed that this interpretation needs to be revised throughout the paper (including title and Abstract).

3) There were several methodological/statistical concerns that need to be addressed.

3.1) The low number of subjects (N = 16 – 1 = 15) may call into question the generalizability of the findings. It is relieving that the univariate analysis largely replicates McGuire et al., 2014, but the MVPA analyses are novel and it is hard to assess their generalizability and the potential of false- positive findings. At the very least, this need to be acknowledged in the Discussion.

3.2) The second condition under which trials were removed from analysis is unclear (subsection “Behavior analysis”). What was the exact criterion to identify these trials? Which error magnitude was still acceptable? Did this mean that, after trial removal, large error magnitudes were always and only associated with change points, and does this bias the results in favor of the author's hypothesis?

3.3) The uncorrected threshold of p < 0.005 is prone to false positives (Eklund et al., 2016), and the authors should use p < 0.001 to define clusters, as suggested in that paper.

3.4) Please clearly describe how "balanced accuracy" was computed, and perform random permutations to determine empirical chance levels and use those as baselines.

4) It was unclear why the RB model was used in this study that need to be addressed.

4.1) The two factors in the experimental design do not seem to map directly onto hazard rate (H) and noise level (K). Indeed, H and K appear to be correlated: in a fast change environment, H should be high by definition, K also needs to be high in order to account for frequent changes of reward target.

4.2) Figure 3—figure supplement 1B: The RU values in the unstable condition is lower than those in the high-noise condition. This is opposite to what is shown in Figure 3B. In other words, this result shows a discrepancy between behavioral data-derived model output and output from true model parameters, which may suggest that the model does not account for the behavior as expected.

4.3) The current model-based behavioral results do not support a direct relationship between model parameters and behavior; instead, the authors just show that both model parameters and behavior change as a function of experimental conditions.

4.4) The model-based fMRI results are mostly replications of previous studies and are not linked to the key results (e.g. Figures 4-6). That is, it is unclear what the model really adds to the conclusions of this paper.

4.5) Given these concerns, reviewers suggested to move model-related results to the supplementary materials.

---

## [Author Response]

Essential revisions:1) Results in Figure 6 are interesting as they show that behavioral adjustments can be predicted from brain activity, replicating the findings by Hampton and O'Doherty, 2007. However, it is not clear that they can be considered "context-dependent". This analysis is based on one condition only, and it is therefore not clear whether or not these signals are context-dependent. The reviewers understand that because there are no ambiguous errors in the unstable context the behavioral adjustment analysis is not possible in this context, and thus, no comparisons across contexts can be performed. The authors can choose to keep these results in the paper, but they should not refer to them as "context-dependent" anywhere in the manuscript, including the title and Abstract.

We have removed all references to the activity predictive of behavioral adjustments being “context-dependent”.

2) The experimental design uses noise level and volatility to define contexts. This is potentially problematic. First, we do not know whether differences between conditions are driven by noise levels, hazard rate, or both. This needs to be discussed. Second, and more importantly, because the noise level is zero in one condition, one context is deterministic whereas the other is probabilistic. This fundamentally changes the meaning of errors and error magnitude between the contexts, such that there is no meaning to error magnitude in the unstable condition (in the sense that larger errors would be any different from smaller errors). Instead, it is a nominal signal that informs subjects of what the new target is, such that an error of 1 carries the same information as an error of 6. Why would such a signal be encoded in a ordinal or linear fashion as in the high noise condition, where the magnitude of the error is meaningful? Thus, given that contexts are defined by whether or not error magnitude is meaningful, is it really surprising that the brain responds differently to error magnitude? Would the authors expect to find the same results if context was defined by a variable that does not directly affect the meaning of error magnitude (e.g. learning about target locations in different environments)? This boils down to the question of whether these are context-dependent errors, or just fundamentally different errors. Given these issues, reviewers felt that describing the results as "context-dependent" is slightly misleading. To be clear, reviewers feel that the findings are important and interesting, but they agreed that this interpretation needs to be revised throughout the paper (including title and Abstract).

We thank the reviewers for raising these issues. We have changed “context-dependent” to “task-dependent” throughout the paper and have substantially revised our interpretations of these findings. For example:

“Second, in this study, we created two qualitatively different task conditions by manipulating both the noise levels and hazard rates. […] Future studies can manipulate hazard rate and noise independently to examine their independent contributions to adaptive learning.”

3) There were several methodological/statistical concerns that need to be addressed.3.1) The low number of subjects (N = 16 – 1 = 15) may call into question the generalizability of the findings. It is relieving that the univariate analysis largely replicates McGuire et al., 2014, but the MVPA analyses are novel and it is hard to assess their generalizability and the potential of false-positive findings. At the very least, this need to be acknowledged in the Discussion.

We now note:

“Although we control the false-positive rates through permutation tests that have been validated empirically (Eklund, Nichols and Knutsson, 2016), it is possible that we lacked the statistical power to detect some effects, and so null results should be interpreted with caution.”

3.2) The second condition under which trials were removed from analysis is unclear (subsection “Behavior analysis”). What was the exact criterion to identify these trials? Which error magnitude was still acceptable? Did this mean that, after trial removal, large error magnitudes were always and only associated with change points, and does this bias the results in favor of the author's hypothesis?

We added the following clarification:

“All of the belief updating models we tested predict that participants’ choice should be along the shortest distance between the previously chosen target and the previously rewarded target. […] Further, this behavior might suggest that participants had lost track of the most recently chosen or rewarded targets.”

Because so few trials were removed, this procedure had little or no influence on the current results.

3.3) The uncorrected threshold of p < 0.005 is prone to false positives (Eklund et al., 2016), and the authors should use p < 0.001 to define clusters, as suggested in that paper.

Eklund et al., 2016, show that non-parametric permutation tests empirically control the false positive rates at nominal levels, and this is true for both of the cluster-defining thresholds they examined (uncorrected voxel *p*<0.01 and uncorrected voxel *p*<0.001). Note that we have corrected in the revised manuscript that the threshold we used of *t*=2.6 corresponds to an uncorrected voxel *p*<0.01. As the Abstract of Eklund et al., 2016, states, “By comparison, the nonparametric permutation test is found to produce nominal results for voxelwise as well as clusterwise inference.” See also their Figure 1. That is, they do not find that a lower cluster-forming threshold is prone to false positives when combined with non-parametric methods of cluster-wise inference, in contrast to their findings for parametric methods of cluster-wise inference.

In response to this comment, though, we did examine how the cluster-forming approach impacted our results, and we report these findings in Figure 5—figure supplement 1. We evaluated two alternative approaches to thresholding: threshold-free cluster enhancement and a cluster-forming threshold of *p*<0.001. Note that in both cases we use non-parametric cluster-wise inference through permutation testing to control the false-positive rates. These two methods yield clusters that are larger (threshold-free cluster enhancement) or smaller (cluster-forming threshold of *p*<0.001) than the approach we use in the main manuscript (cluster-forming threshold of *p*<0.01). Given that the appropriate interpretation of observing a significant cluster is that there is a significant effect somewhere within that cluster, these two methods therefore yield greater (cluster-forming threshold of *p*<0.001) or lesser (threshold-free cluster enhancement) spatial specificity than the approach in the main manuscript. In general, both methods yield similar main results, with a loss of spatial specificity when using threshold-free cluster enhancement and some loss of sensitivity (one of three effects does not remain significant) when using a cluster-forming threshold of *p*<.001. We further stress that, based on the results of Eklund et al., 2016, none of these three methods is more or less conservative than the others, because they all control the cluster-wise false positive rate at *p*<0.05. Where they differ is in their spatial sensitivity/specificity.

3.4) Please clearly describe how "balanced accuracy" was computed, and perform random permutations to determine empirical chance levels and use those as baselines.

We added the following sentences to clarify balanced accuracy:

“For balanced accuracy, we first calculated the classification accuracy within each category, and then averaged the accuracies across all categories. […] The baseline accuracy for two categories was 50% and for three categories was 33%.”

4) It was unclear why the RB model was used in this study that need to be addressed.4.1) The two factors in the experimental design do not seem to map directly onto hazard rate (H) and noise level (K). Indeed, H and K appear to be correlated: in a fast change environment, H should be high by definition, K also needs to be high in order to account for frequent changes of reward target.

We manipulated these two factors, hazard rate and noise level, to create two qualitatively different conditions where participants should use small errors differently. In the reduced Bayesian model, the parameters *H* and *K* represent a participant’s subjective estimates of the hazard rate and noise level in a given condition, respectively. Fits of the reduced Bayesian model indicated that participants differentiated between hazard rates but not noise levels across the two conditions. Though *K* is higher than its objective value in low noise/high hazard condition, it did not need to be so in order to account for the frequent changes of reward target. Both *H* and *K* influence both CPP and RU values, which in turn influence choices to switch targets.

4.2) Figure 3—figure supplement 1B: The RU values in the unstable condition is lower than those in the high-noise condition. This is opposite to what is shown in Figure 3B. In other words, this result shows a discrepancy between behavioral data-derived model output and output from true model parameters, which may suggest that the model does not account for the behavior as expected.

The “behavioral data-derived model output” was based on fits to the behavioral data, with subjective estimates of noise level and hazard rate determined by free parameters. This model thus was fully capable of accounting for behavior of subjects using the true, objective values (i.e. the free parameters could have settled on those values). However, consistent with our earlier results (e.g. Nassar et al., 2010; Li et al., 2019), that is not what we found here. Instead, behavior tended to follow principles of the normative model but based on subjective estimates of those parameters that tended to differ from their objective values (using formal model selection, models with both free parameters were superior to models using objective values for both hazard rate and noise level, and to models using just objective values for noise level and just hazard rate as a free parameter).

Note that, consistent with the suggestions below, both of these figures are now provided (behavioral predictions and neural activations using subjective estimates in Figure 2—figure supplement 3 and true values in Figure 2—figure supplement 4) and the corresponding section of the text has been shortened.

4.3) The current model-based behavioral results do not support a direct relationship between model parameters and behavior; instead, the authors just show that both model parameters and behavior change as a function of experimental conditions.

It is not true that we “just show that both model parameters and behavior change as a function of experimental conditions.” The model was fit directly to behavioral data. Based on these fits, we show that the fitted reduced Bayesian model accounts for participants’ behavior well across different conditions. We also used formal model-selection procedures (BIC) to show that the reduced Bayesian model accounted for behavior better than: 1) an approximately ideal observer that uses the true values of noise level and hazard rate; 2) a model that assumes a fixed learning rate in each condition; and 3) models that supplement the reduced Bayesian model with other features. The contribution of what is now Figure 2—figure supplement 3 is to describe how the reduced Bayesian model accounts for task-dependent behavioral responses to errors (i.e. the influence of error magnitude and error history in the noisy condition), by showing how the key internal representations that govern the behavior of the model (CPP and RU) change as a function of experimental conditions.

4.4) The model-based fMRI results are mostly replications of previous studies and are not linked to the key results (e.g. Figures 4-6). That is, it is unclear what the model really adds to the conclusions of this paper.4.5) Given these concerns, reviewers suggested to move model-related results to the supplementary materials.

We agree that the primary value of the model-based fMRI results is to demonstrate that we replicate activations we observed in a previous study with the new task used in this study. Though we think reporting such replication is important, we do not want to distract from the key results of this paper. We therefore have moved the model-based fMRI results to Figure 2—figure supplements 3 and 4.